# BN-Embedded Perylenes

**DOI:** 10.3390/molecules26237148

**Published:** 2021-11-25

**Authors:** Xiangdong Fang

**Affiliations:** College of Chemical Science and Engineering, Tongji University, 1239 Siping Road, Shanghai 200092, China; xdfang@tongji.edu.cn

**Keywords:** perylene, BN-aromatics, PAHs, palladium catalysis, coupling

## Abstract

Transition metal catalyzed coupling reaction strategy has been utilized in the synthesis of two novel BN-perylenes starting from halogenated BN-naphthalene derivatives. The molecular structures and packing modes of BN-perylenes were confirmed by NMR spectroscopy and X-ray single-crystal diffraction experiments. Their photophysical properties were further investigated using UV-vis and fluorescence spectroscopy and DFT calculations. Interestingly, the isosteric BN-insertion in perylene system resulted in stronger π-π stacking interaction both in solid and solution phases. The synthesized BN-perylenes are proved to be highly stable and thus provide a new valuable platform for novel organic materials applications which is otherwise inaccessible to date.

## 1. Introduction

The advent of “the organic era” has gradually evolved and fostered to enrich the understanding and development at molecular levels in next-generation research areas of carbon materials, such as organic photo-catalysis, organic optoelectronics, organic photonics, organic spintronics as well as organic superconductors [1,2,3,4,5,6,7]. Organic materials are considered to be potentially advantageous over inorganic counterparts, owing to their great reputes in tailor-made molecular structures and low-cost device fabrication process. Highly rigid conjugate systems (e.g., polycyclic aromatic hydrocarbons, known as PAHs) have attracted prodigious interest with remunerative applications in organic optoelectronic devices [3,4,5,6,7]. Perylenes, as an example of two-dimensional graphene planar segments, demonstrate unique structure-dependent properties and perylene diimides (PDI) with n-type semiconductor characteristics have been successfully commercialized in organic solar-cell devices (Figure 1) [8,9,10,11,12,13,14,15]. At the required level of practical verisimilitude, structural tuning with heteroatom substitution of sp^2^-hybridized carbons becomes particularly attractive with an aim to exquisitely control the electronic properties of carbon-rich conjugate systems [16,17,18,19,20]. In this respect, the isosteric substitution of C=C bond with B-N bond is used to influence the electronic density and energy levels of corresponding frontier molecular orbitals (e.g., HOMOs and LUMOs) and hence to modulate molecular properties such as band gap, optoelectronic and catalytic properties [21,22,23]. However, the syntheses of BN-substituted PAHs are still in a dormant phase with only sporadic reports and the study of BN-PAHs cannot be further sustained for lack of enumerating stable BN-aromatic building blocks, albeit this contradiction in reality is a shock of startling [24,25]. Furthermore, the paramount challenges also come from the competence for orderly implementation of BN dipoles into the conjugate systems rather than some random choices, to precisely control both the orientation and quantity of the inserted BN dipoles.

The structure of perylene, benefiting from its naphthalene-fused π-delocalization, can be easily modified and well-utilized in organic functional materials [26]. Introducing various functional groups into the bay region of perylene molecule as well as PDI’s imide sites, has led to the discovery of a plethora of organic functional materials with novel properties. However, this peripheral modification possesses a less pronounced impact on the intermolecular interactions such as π-π co-facial crystal stacking, which are important in affording high charge-carrier transporting property in organic semiconductors [27,28,29]. Therefore, it is envisioned that embedding BN dipoles in the core structure of perylene not only alters the HOMO-LUMO band gap, but also gives rise to stronger intermolecular interactions and densely packed molecular π-stackings, which are auspicious for high mobility in charge transport. In addition, it is important that designed BN-perylenes should be inherently stable enough for further device applications. Most recently, Wagner strategically developed a Wacker-type (Ph_3_P)Au(NTf_2_)-catalyzed cyclization of corresponding amine and alkyne terminals for the synthesis of substituted (BN)_2_-perylene derivatives [30]. However, a deteriorated π-conjugation system has been observed in this case, as indicated by noticeable BN bond elongation and twist away from the planar structure. Herein, we present the first synthesis of BN-substituted perylenes **1** and **2** by 4a,8a-azaboranaphthalene (ABN) building block, in which BN units are placed at the junction position of naphthalene ring (Figure 1) [31]. Both compounds (**1** and **2**) demonstrate the characteristic of planar structures with interesting π-π co-facial stackings, as evidenced by the X-ray single crystal diffraction experiments.

## 2. Results

ABN dibromide **3b** has been synthesized by our group, [32] whose ready availability provides a potential building block for BN-PAHs syntheses. We began our investigation by treating a mixture of dibromide **3b**, COD (COD = 1,5-cyclooctadiene) and bpy (2,2′-bipyridyl) in the presence of Ni(cod)_2_ complex in THF under the Yamamoto coupling protocol (1.0 equiv. of Ni(cod)_2_/COD/bpy per C−Br unit) (Figure 1), [33] with an expectation of the favorable six-membered ring formation. We obtained a highly selective formation of (BN)_2_-perylene **2** in a nearly quantitative yield, which could be reproduced regardless of temperature variation. In a similar route, **2** could be synthesized from ABN dichloride **3a** in 63% yield. Encouraged by this preliminary result, our attention has turned to the preparation of BN-perylene **1** with different naphthalene units. Suzuki cross-coupling of dibromide **3b** and 1-naphthylboronic acid provided **4** in 56% yield, whose identity has been confirmed by ^1^HNMR and X-ray diffraction experiments. A simple switch of the solvent from THF/H_2_O to anhydrous THF could easily resolve the plague of reduced side-product **5** in these coupling reactions. Compound **4** then underwent the next intramolecular cyclization with the help of Pd(OAc)_2_/Xantphos and K_2_CO_3_ in the presence of 1,4-dioxane at 110 °C to afford the desired BN-perylene **1** product in 50% yield. In this step, the addition of anhydrous LiCl was crucial to the final success of the cyclization, because it could effectively avoid forming undesired **5** (see reference [9]) and promote the formation of product **1** [34].

## 3. Discussion

Single crystals of BN-perylene **1** (yellow plates) and (BN)_2_-perylene **2** (yellow-green plates) suitable for X-ray diffraction experiments could be obtained by slow evaporation of their toluene solutions under ambient conditions. Like perylene, a completely planar structure with sandwich herringbone stacking arrangements in the crystal lattice was observed in both cases of **1** and **2**. However, the molecular packings of **1** and **2** differ from perylene in a subtle yet pronounced way (Figure 2). First, π-stacking distances in BN-perylene **1** (3.388 Å) and (BN)_2_-perylene **2** (3.380 Å) are distinctively shorter than that observed in perylene (3.458 Å)^26^, which is consistent with strong π-interaction between BN-aromatic rings induced by BN-dipoles. Secondly, albeit the C−H⋯π interactions in perylene and (BN)-perylene **1** share a similar structural characteristic of bay-area stacking in the unit cells, (BN)_2_-perylene **2** molecules tend to stack together with *peri* C−H bonds. This may reflect in a different Coulombic demand by the BN-dipole engagement in the system. It is further noted that a shorter interaction distance (2.684 Å) by *peri* C−H⋯π interaction of **2** is recorded in comparison to those (3.302~3.526 Å) of bay C−H⋯π interactions in **1** and perylene. The bond distances of the C−C bond bridging two naphthalene rings in **1** (avg. 1.475 Å) and **2** (avg. 1.478 Å) are approximately identical with that (avg. 1.474 Å) in the perylene molecule, suggesting a similar π−π interaction between two naphthalene units. All structural metric parameters of **1** and **2** are consistent with those of the BN-aromatic molecules, except that the internal bond angles of the bridging six-membered rings in **1** (C−C−C: ca. 117.4°~117.6°) and **2** (B−C−C: ca. 117.1°~117.4°) become noticeably smaller than the ideal *sp*^2^-hybridized bond angle (120°).

The UV absorption and emission spectra of **1** and **2** in THF were subsequently studied to further reveal their structural details (Figure 3). The UV absorption peaks in the range of 350~450 nm are likely to be the characteristic of benzene-ring absorption with the fine structures, which is apparently diminishing when the naphthalene units are gradually replaced by the ABN counterparts. In this respect, (BN)_2_-perylene **2** demonstrates a very weak absorption in this region; while all-carbon perylene with four benzene motifs maximizes its B-band absorption peaks in the same range. Both BN-perylenes **1** and **2** possess strong absorption peaks in the range of 270 to 350 nm, which may be related to the π→π* electron transition of the conjugated alkene systems, owing to less aromatic character of BN-aromatics in nature. In this respect, the absorption peak of **2** is red-shifted of roughly 25 nm relative to that of **1**, consistent with further decreased aromaticity in the (BN)_2_-substituted perylene system. For comparison, the UV absorption spectrum of 1,3-cyclohexadiene is also provided (dotted line in Figure 3A), which suggests that the absorption peaks below 270 nm could be attributed to localized diene systems. Despite similar fine-structure patterns of their emissive spectra, the emission peaks of BN-substituted **1** and **2** are clearly broadened relative to those of perylene (Figure 3B). In particular, a bathochromic shift of 25 nm in wavelength has been observed for (BN)_2_-perylene **2** compound relative to **1**, which is also solvent-independent as switching from THF to toluene.

In principle, the structural rigidity of perylene, **1**, and **2** with such delocalized π-systems will induce a phenomenon known as aggregation-caused quenching (ACQ), [35] which prompted us to reason that favorable π-π stacking interaction in the crystal structure of **1** and **2** stimulate the ACQ in their liquid phases under certain experimental conditions. To verify this hypothesis, we designed an ACQ experiment by utilizing the hydrophobic property of these molecules and good solubility in water-miscible solvents (Figure 4). The addition of water reduced the solubility of BN-perylene molecules and promoted their subtle π-stacking aggregations, as demonstrated by quenching of the fluorescence. To our satisfaction, the intensities of the fluorescence peaks reached the summits at distinctive H_2_O volume fractions in THF (perylene: ca. 60 vol%; **1** and **2**: ca. 40%) followed by characteristic declining slopes respectively, in agreement with an increased π-π interaction observed in the structures of BN-perylenes **1** and **2**.

To elucidate the structural and electronic effects of BN-substitution in the perylene system, DFT calculations with BN-perylenes (**1** and **2**) and perylene were conducted using the B3LYP/6-31G* and M062X/6-31+G(d) levels of theory, respectively. Both calculation results show that the entire frameworks of BN-perylenes possess a highly planar geometry, which correlates well with those of the X-ray diffraction analysis. In further comparison, DFT calculations at M062X/6-31+G(d) level revealed the weak interactions between BN-perylene molecules leading to predict the formation of a dimeric unit in both **1** and **2**; while the B3LYP/6-31G* calculations failed, which prompted us to choose the M062X/6-31+G(d) level in the final calculation. The HOMO and LUMO contours along the entire π-annulated frameworks in **1**, **2** and perylene are similar and the energy levels of the HOMO and LUMO orbitals in all three cases vary in relatively small discrepancies (Figure 5). In this respect, the unexpected bathochromic shift in the emission study of **2** (Figure 3B) may be rationalized by a better conjugation of ABN units, which generates a smaller HOMO-LUMO bandgap owing to the smaller repulsion between the HOMO and LUMO electrons in the excited state. The nucleus-independent chemical shift (NICS) calculation has been used to provide the information about the aromaticity of five fused rings in **1**, **2** and perylene (Figure 5). Interestingly, the aromaticity of the naphthalene ring and the BC_5_/B_2_C_4_ rings in **1** and **2** has been remarkably affected by the BN insertion in the perylene system, indicating an effective electronic conjugation in the whole BN-perylene molecules. In this way, the NISC(1)_ZZ values of the central rings vary from 2.540 (perylene) to 2.145 (**1**, less antiaromatic character) and 3.218 (**2**, more antiaromatic character); whereas those NICS(1)_ZZ values of the peripheral rings change from −9.133 (perylene) to −8.644/−6.349 (**1**) and −5.941 (**2**) with decreased aromaticity.

## 4. Materials and Methods

All air- and moisture-sensitive reactions are conducted with magnetic stirring in oven-dry glassware under nitrogen atmosphere using anhydrous solvents and standard Schlenk-line techniques with an MBraun Labstar-MB10 glovebox. The oxygen and moisture levels in the MBraun glovebox were constantly monitored by both oxygen and moisture analyzers to ensure O_2_/H_2_O levels were below 0.1 ppm. Diethyl ether, tetrahydrofuran, and hexane are dried and distilled by treatment with sodium metal slices under a nitrogen atmosphere. Dichloromethane is dried and distilled by treatment with calcium hydride under a nitrogen atmosphere. Chemicals were purchased from chemical suppliers and used without further purification unless otherwise specified. Flash or thin-layer column chromatography was performed over silica gel (40–60 ¦Ìm) purchased from Yantai Jiangyou Co., China. ^1^H, ^13^C, ^11^B NMR spectra (Appendix A) were collected on a Bruker ARX400 400 MHz NMR spectrometer using residue solvent peaks as an internal standard (^1^H NMR: CDCl_3_ at δ 7.26 ppm; ^13^C NMR: CDCl_3_ at δ 77.00 ppm; ^11^B NMR: BF_3_⋅Et_2_O at δ 0.0 ppm). Data for ^1^H NMR were recorded as follows: chemical shift (δ, ppm), multiplicity (s = singlet; d = doublet; t = triplet; m = multiplet; bs = broad singlet), coupling constant (*J* in Hz), integration. Mass spectra were collected at Bruker microTOF II ESI-TOF and Waters Micromass GCT Premier-TOF mass spectrometers. The UV-Vis and fluorescence spectra were collected on an Agilent 8354 and a Hitachi F-7000 spectrophotometer respectively. The X-ray single-crystal diffraction experiments were performed with a Bruker D8 VENTURE X-ray diffractometer with CMOS detector.

### 4.1. Synthesis of 4-bromo-5-(1-naphathyl)-4a,8a-azaboranaphthalene

4,5-dibromo-4a,8a-azaboranaphthalene **3b** (28.6 mg, 0.1 mmol), 1-naphthylboronic acid (34.4 mg, 0.2 mmol), K_2_CO_3_ (27.7 mg, 0.2 mmol), and Pd(dppf)Cl_2_ (3.6 mg, 0.005 mmol) were loaded in a Schlenk flask, which was then evacuated and recharged with nitrogen in three times. THF (3 mL) was injected with a syringe. The mixture was heated 12 hr at 110 ℃. The solvent was removed by vacuum and the residue was extracted with CH_2_Cl_2_/H_2_O twice. The combined organic layers were washed with brine and dried with anhydrous MgSO_4_. The solution was concentrated and further purified by flash column chromatography (hexane:dichloromethane = 10:1). Product **4** was obtained as a light-yellow solid (37.3 mg, 56%). CCDC deposition number: 2107102.

^1^H NMR (400 MHz, CDCl_3_): δ 7.97–7.79 (m, 5H), 7.64-7.51 (m, 3H), 7.46 (t, *J* = 7.0 Hz, 1H), 7.32 (t, *J* = 6.3 Hz, 2H), 6.90 (t, *J* = 6.9 Hz, 1H), 6.59 (t, *J* = 7.1 Hz, 1H). 

^13^C NMR (101 MHz, CDCl_3_): δ 143.2, 141.8, 140.2, 134.1, 133.8, 133.4, 126.7, 126.2, 125.4, 114.3, 113.9.

^11^B NMR (193 MHz, CDCl_3_): δ 26.99.

HRMS (EI+, 70 eV): calculated for C_18_H_13_^10^BBrN [M]^+^: 332.0361; found: 332.0370.

### 4.2. Synthesis of 3a-aza-9a-bora-perylene

4-bromo-5-(1-naphthyl)-4a,8a-azaboranaphthalene (**4**) (33.4 mg, 0.1 mmol), LiCl (4.3 mg, 0.1 mmol), K_2_CO_3_ (27.6 mg, 0.2 mmol), Pd(OAc)_2_ (2.3 mg, 0.005 mmol), and Xantphos (5.8 mg, 0.01 mmol) were loaded in a Schlenk flask, which was then evacuated and recharged with nitrogen three times. 1,4-dioxane (5 mL) was injected with a syringe. The mixture was heated for 12 hr at 130 ℃. The solvent was removed by vacuum and the residue was extracted with CH_2_Cl_2_/H_2_O twice. The combined organic layers were washed with brine and dried with anhydrous MgSO_4_. The solution was concentrated and further purified by thin-layer column chromatography with hexane/CH_2_Cl_2_ (1:1) as eluent. Product **1** was obtained as a light-yellow solid (12.6 mg, 50%). The single crystals of **1** suitable for X-ray diffraction were cultivated from its CH_2_Cl_2_ solution. CCDC deposition number: 2107101.

^1^H NMR (400 MHz, CDCl_3_): δ 8.31 (d, *J* = 7.3 Hz, 2H), 8.24 (d, *J* = 6.9 Hz, 2H), 7.81 (d, *J* = 7.9 Hz, 2H), 7.75 (d, *J* = 7.9 Hz, 2H), 7.52 (t, *J* = 7.7 Hz, 2H), 6.87 (t, *J* = 6.8 Hz, 2H). 

^13^C NMR (101 MHz, CDCl_3_): δ 135.6, 134.4, 131.1, 129.6, 128.9, 127.7, 125.7, 120.9, 115.4.

^11^B NMR (193 MHz, CDCl_3_): δ 29.00.

HRMS (EI+, 70 eV): calculated for C_18_H_12_^10^BN [M]^+^: 252.1099; found: 252.1102.

### 4.3. Synthesis of 3a,9a-diaza-3a^1^,6a^1^-dibora-perylene

A mixture of 1,5-cyclooctadiene (COD, 540 mg, 5 mmol) and 2,2′-bipyridine (Bpy, 390 mg, 2.5 mmol) in a Schlenk flask was dissolved in 30 mL of THF in the glovebox. Ni(COD)_2_ (680 mg, 2.5 mmol) was added with magnetic stirring into the above colorless solution to afford a deep purple-blue solution, which was further stirred 5 min at room temperature. A solution of 4,5-dihalo-4a,8a-azaboranaphthalene (**3a**, 97 mg, 0.49 mmol; **3b**, 140 mg, 0.49 mmol) in 10 mL of THF was added into the above purple-blue solution. The sealed Schlenk flask was taken out of the glovebox and the reaction mixture was stirred 12 hr at room temperature. The solvent was removed under vacuum and the residue was extracted with refluxing toluene (30 mL). After filtration and removal of toluene under vacuum, product **2** was isolated as a light-yellow solid (from **3a**: 39 mg, 63%; from **3b**: 62 mg, 100%). The single crystals of **2** suitable for X-ray diffraction were cultivated from its toluene solution. CCDC deposition number: 2107103.

^1^H NMR (400 MHz, CDCl_3_): δ 8.05 (d, *J* = 6.8 Hz, 4H), 7.59 (d, *J* = 6.8 Hz, 4H), 6.71 (t, *J* = 6.8 Hz, 4H). 

^13^C NMR (101 MHz, CDCl_3_): δ 132.0, 127.9, 115.0.

^11^B NMR (193 MHz, CDCl_3_): δ 30.36.

HRMS (EI+, 70 eV): calculated for C_16_H_12_^10^B_2_N_2_ [M]^+^: 252.1259; found: 252.1252.

### 4.4. Determination of Structures of 1, 2, and 4

The single crystals of **1**, **2**, and **4** were immersed in FOMBLIN oil (HVAC 140/13, Sigma-Aldrich), mounted on a glass fiber, and examined on a Bruker D8 VENTURE diffractometer equipped with an Oxford Cryostream 800 low-temperature device using a nickel-filtered Cu K¦Á radiation source (λ = 1.54178 Å) and a Bruker PHOTON II detector at 150 K and 298 K, respectively. All data were integrated with SAINT and a multi-scan absorption correction using TWINABS was applied. The structures were solved by direct methods using SHELXT and refined by full-matrix least-squares methods against *F*^2^ by SHELXL-2018/1. All non-hydrogen atoms were refined with anisotropic displacement parameters. The hydrogen atoms were refined their pivot atoms for terminal sp^3^ carbon atoms and 1.2 times for all other carbon atoms.

### 4.5. Density Function Theory (DFT) Calculations

Density functional theory (DFT) calculations were carried out with Gaussian 09 software package. The geometry optimization and frequency calculations were carried out at M062X/6-31G+(d) level.

## 5. Conclusions

In summary, we have provided effective synthetic protocols for the incorporation of BN units into the perylene system, which affects their π-stacking mode and corresponding photophysical properties. These novel BN-perylene molecules exhibit highly planar structures, unique delocalized backbones, and good compound stability. More importantly, it is demonstrated that the π-stacking interaction between perylene systems may be engineered by controlling the number of BN units in the perylene molecule. These results suggest that BN-embedded PAHs are promising candidates for novel organic electronic applications. Further studies to better understand the chemistry of BN-perylenes and extend their utilities in novel organic electronic materials are currently underway in our laboratory.

## Data Availability

Not applicable.

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
