# Peer review of "BN-Embedded Perylenes"

_molecules, 2021, doi:10.3390/molecules26237148_

Round 1

Reviewer 1 Report

see attached

Author Response

Thanks for the opinions of the reviewer.

1) The lower yields of BN-perylene (1) is perhaps owing to a competing side reaction of 5-membered ring formation. Although it has been observed by NMR spectroscopy. However, its single-crystal structural conformation has not been achieved. So the reaction mechanism was not further discussed in the main text.

2) The compounds are all characterized by NMR and HRMS, which are generally accepted by most organic journals. The melting point data will be disclosed in a full article in the near future.

3) I will certainly consider citing the reference paper by Guan et al.

4) My synthetic method was at room temperature, because Ni(COD)2 is reactive enough to initiate the reaction without heating. The preparative route used in ref 13 was heated, because the standard Yamamoto coupling was conducted by heating the mixture at 70-80oC. Furthermore, it was likely that the authors of ref 13 did not even attempt to try the room-temperature condition. Therefore, it is not necessary to discuss this difference in the main context of the paper.

Reviewer 2 Report

The paper from Fang describes synthesis and characterization of new B-N aromatic molecules. The paper also shows the electronic properties of such molecules. The paper is very interesting because describes also the Fluorescence of the compounds which In my opinion gives more information on the electronic states of the molecules containing the B-N bonds For all the reasons reported here the paper should be published in Materials because gives a lot of new data on aromatic compounds bearing perturbation on the orbitals due to the presence of B-N moiety.

The paper is very interesting to me. I found the science well described and suitable for further improvements. 

I think that the paper should be published as is.

Author Response

Thanks for the opinion of the reviewer.

Reviewer 3 Report

The contents described in this paper are found be the essentially same as that cited by the author in the ref #13 (ACIE 2021, 60, 23313) and is therefore unsuitable for publication, although the foregoing paper doesn't include the compound 1. The synthetic methods presented in the paper, which is the appealing point of this paper, are exactly the same as that of the previous study, giving no original achivement. Thus, this paper cannot be published in any media including this journal. We therefore cannot help recommending this paper to be rejected, unless new attractive findings are to be added.

Author Response

Thanks for the reviewer.

Two methods that you cited are NOT exactly same. My synthetic method was conducted at room temperature, and the method by ref #13 was reported at refluxing temperature of THF (ca. 70oC). Furthermore, the yields were different (~100% in my case, ~80% in ref #13). Finally, dichloro BN-naphthalene was also reported in my manuscript to give (BN)2-perylene product in good yields, which was not even mentioned in ref #13.  

Additionally, dibromo and dichloro BN-naphthlenes were first reported from my group.

Reviewer 4 Report

Manuscript ID: Manuscript ID: molecules-1430771

Type of manuscript: Article
Authors: X. Fang.

The paper reports on novel and valuable results on synthesis of novel substituted perylenes and on their characterization using a series of experimental techniques and using DFT calculations at B3LYP/6-31G* and M062X/6-31+G(d) levels. The paper provides particular results on synthetic protocols for the incorporation of BN units into perylene system. It was shown that Pi-stacking interaction in a BN-perylene solid may be influenced by tuning the number of BN units in the perylene molecule.

As to reviewer comments, questions and/or suggestions there is as follows.

  1. The manuscript brings up results on structuring of BN-perylenes within a solid (in Fig.2) including plane-to-pane distance and followed by assumptions on possible ways of Pi-stacking. The DFT calculations presented in the paper concern separate molecules (see for example Fig. 5). So the orbital energies calculated, including the frontier orbital energies, are not affected by the electronic system of the neighboring molecules. Nontheless, the DFT calculations using Gaussian software are well known as highly valuable also for model organic solids, but certain energy scaling procedures are to be applied to molecular orbital energy values before comparing them to energy bands within an organic solid. One of possible energy scaling procedures is linear scaling which would treat separately pi type and sigma type orbitals in valence band, as well as pi* and sigma* orbitals in conduction band [refs. 2-4 in the list below]. Discussion on this subject should be expanded in the manuscript in order to avoid misleading of a reader. Enormous number of papers is devoted to this subject. Below I would just point on some refs. which I found as examples.

  1. G. Hill, A. Kahn, J. Cornil, D.A. dos Santos, J.L. Bredas, Occupied and unoccupied electronic levels in organic pi-conjugated molecules: comparison between experiment and theory, Chem. Phys. Lett. 317 (2000), 444-450.
  2. A.S. Komolov, E.F. Lazneva, N.B.Gerasimova, Y.A. Panina, A.V. Baramygin, G.D. Zashikhin, S.A. Pshenichnyuk, Structure of vacant electronic states of an oxidized germanium surface upon deposition of perylene tetracarboxylic dianhydride films, Physics of the Solid State 58, (2016) 377.
  3. A.S. Komolov, E.F. Lazneva, N.B.Gerasimova, Y.A. Panina, V.S. Sobolev, A.V. Koroleva, S.A. Pshenichnyuk, N.L. Asfandiarov, A. Modelli, B. Handke, O.V. Borshchev, S.A. Ponomarenko, Conduction band electronic states of ultrathin layers of thiophene/phenylene co-oligomers on an oxidized silicon surface, J. Electron Spectr. Rel. 235 (2019) 40.
  4. M. Scheer, P.D. Burrow, pi* Orbital System of Alternating Phenyl and Ethynyl Groups: Measurements and Calculations, J. Phys. Chem. B 110 (2006) 17751-17756.

I recommend this manuscript for publication after minor revision.  

Regards, Reviewer

Author Response

Thanks for the reviewer's valuable feedbacks.

I will certainly consider adding some of these papers in the reference part.

Round 2

Reviewer 3 Report

Compared with the results included in ref # 13, the differences for the reaction conditions and the product yield, which the author claims, are trivial, which I understood to be essentially the same. Dichloro BN-naphthalene only serves as a synthetic intermediate, which cannot warrant any originality of the research subject. I believe that the rebuttal by the author is not valid and cannot therefore be published for any reason.